# OpenReview forum: "PATTERNS AND MECHANISMS OF CONTRASTIVE ACTIVATION ENGINEERING"
_ICLR.cc/2025/Workshop/BuildingTrust — BuildingTrust_

### Official Review · Reviewer_wFNu · 2025-02-16

**Rating:** 7
**Confidence:** 4

**Review:**

# Strengths

* **Relevant Topic**: Controlling LLM behavior at inference time is a crucial area of research for safety and alignment, and CAE has been a more promising approach.

* **Empirical Investigation**: Paper attempts a comprehensive systematic investigation of CAE, varying parameters like dataset size, steering strength, and model size.

* **Meaningful Validation Of Previous Work**: Sweeps across layers and steering strengths.

* **Focus on Out-of-Distribution (OOD) Generalization**: The paper explicitly addresses the crucial question of OOD performance, which is often overlooked. The creation of a new OOD evaluation dataset is a positive step, although the dataset itself needs more scrutiny.

* **Perplexity Analysis**: The attempt to quantify negative side effects of steering via perplexity is a good idea, addressing a critical aspect of controllability.

# Weaknesses

* **Missing Citations**: L32, L34, L81, L108

* **Perplexity Analysis Vagueness**: The perplexity analysis (Section 6) is vague. How was the "subset of texts in the Pile" chosen? What constitutes a "large change" in likelihood?

* **Red-Teaming Results**: The red-teaming experiments are inconclusive.

* **Limited Model Evaluations**: Evaluations limited to Llama Family of Models: Would help if included evaluations for other model families with different architectures (eg Gemma)

# Questions

* See Weaknesses

---

### Official Review · Reviewer_wYBC · 2025-02-27
**Comprehensive CAE Steering Experiments: Confirmation with Limited Novelty or Narrative**

**Rating:** 4
**Confidence:** 4

**Review:**

Summary: The paper presents a wide-range of experiments on CAE steering vectors.
The main experimental findings are:
Section 4. In-Distribution Analysis
Identify optimal layers for steering Llama 8B and 70B by sweeping across layers (average across behaviors of MWE dataset)
Evaluate how many training samples are needed for generate reliable steering vectors.

Section 5. OOD Analysis
Test OOD performance of steering vectors by creating a dataset that has a distributional shift compared to the training dataset.
OOD fails  - would expect left low and right high values. -> confirms findings from Tan et al


Section 6. Perplexity of Steering
Turner et al. find that steering for positive concepts reduces perplexity on positive samples and increases on negative samples..
The paper finds that analogously steering for "French", increases perplexity on Dutch and decreases perplexity on French.

Section 7: Red Teaming Steering vectors with EPO
Paper tests how adversarial inputs generated with EPO generate inputs that make steering vectors act in the opposite direciton.
Which is interesting, but given findings fall short of being conclusive.

Strengths:
1. Good introduction into Contrastive Activation Engineering, relevant literature and methods.
2. Extensive Experiments. The paper runs many different experiments (layer sweeps, OOD, ID, Adversarial examples) that confirm previous findings or slightly extend them. This is valuable to confirm and strengthen existing results.

Weaknesses:
- 1. Marginal contributions, with small benefit over existing work. Most of the findings are in line with previous work, as stated in the paper.
- 2. No cohesive narrative. Findings that are interesting but not investigated in depth enough.
Focus on less results, create a cohesive narrative, expand these results.


Citation and Reference Issues
•	There are several instances of “?” in place of citations. These citation failures should be resolved to ensure proper referencing and clarity.

---

### Official Review · Reviewer_vSFa · 2025-03-02
**Review for "PATTERNS AND MECHANISMS OF CONTRASTIVE ACTIVATION ENGINEERING"**

**Rating:** 4
**Confidence:** 4

**Review:**

**Summary**

This paper provides an empirical study of Contrastive Activation Engineering (CAE) for steering large language models. The authors investigate the effectiveness of CAE, its limitations, and potential negative side effects.

**Strengths:**

- **Comprehensive Empirical Analysis:** The paper conducts a range of experiments, varying dataset size, steering strength, and layers, providing a decent overview of CAE's performance under different conditions. For example, Figures 2-5 show sweeps across layers and steering strengths for different models and dataset sizes. Figure 6 explores the impact of the number of examples used to generate steering vectors.
- **Practical Focus:** The study considers real-world applicability, investigating out-of-distribution performance (Section 5), adversarial robustness (via EPO, Section 7), and the impact on perplexity (Section 6), which are relevant to deployment.
- **Out-of-Distribution Analysis:** The creation of the small evaluation dataset (described in Section 5) represents a useful and novel contribution. The dataset comprises 540 questions, spanning 9 target behaviors, to mimic real-world deployment scenarios.
- **Analysis of Samples for Steering Vectors:** The paper shows that performance improvement from using more samples begins to decrease at 55-89 samples (Figure 6).

**Weaknesses:**

- **Broken Citations:** Many citations are rendered as "?". This makes it difficult to verify the claims. Examples include several citations in the related works (Page 2) like (Turner et al., 2024; ?).
- **Broken Intra-Paper Link:** There is at least one instance of a broken internal link. On page 2, the authors state, "Successfully red-team steering vectors with evolutionary prompt optimization in ??, albeit they're unlikely to be observed naturally." referring to a section that is not properly linked.
- **Lack of Clarity on Choice of Experiments**: The paper doesn't effectively motivate some experimental designs and lacks justification for its parameters. For example, the choice of the Fibonacci sequence for sweeping sample sizes (page 4) seems arbitrary without further explanation.
- **Limited Novelty:** While the empirical study is extensive, the paper doesn't introduce fundamentally new techniques or theoretical insights. The core methodology (contrastive activation addition) is taken directly from prior work (Panickssery et al., 2024, explicitly acknowledged on page 3). The contribution is primarily in the breadth of the empirical evaluation, which while **valuable**, has **diminished impact due to the presentation**.

---

### Decision · Program_Chairs · 2025-03-01

Accept